# SOphrology Intervention to Improve WELL-Being in Hospital Staff (SO-WELL): Protocol for a Randomized Controlled Trial Study

**DOI:** 10.3390/ijerph20021185

**Published:** 2023-01-09

**Authors:** Frédéric Dutheil, Lénise M. Parreira, Bruno Pereira, Maryse Baldet, Frédérique Marson, Christine Chabaud, Magali Blot, Julien S. Baker, Marek Zak, Guillaume Vallet, Valentin Magnon, Maëlys Clinchamps, Senem Altun

**Affiliations:** 1Physiological and Psychosocial Stress, CNRS UMR 6024, LaPSCo, University Clermont Auvergne, WittyFit, 63000 Clermont-Ferrand, France; 2Preventive and Occupational Medicine, University Hospital of Clermont-Ferrand (CHU), 63000 Clermont-Ferrand, France; 3The Clinical Research and Innovation Direction, CHU Clermont-Ferrand, 63000 Clermont-Ferrand, France; 4Pole REUNIRRH, University Hospital of Clermont-Ferrand (CHU), 63000 Clermont-Ferrand, France; 5Pole MobEx (Mobility–Exercise), University Hospital of Clermont-Ferrand (CHU), 63000 Clermont-Ferrand, France; 6Sophrologist (Caycedo Method)–Trained in Neurolinguistic Programming (NLP), University Hospital of Clermont-Ferrand (CHU), 63000 Clermont-Ferrand, France; 7Centre for Health and Exercise Science Research, Department of Sport, Physical Education and Health, Hong Kong Baptist University, Kowloon Tong, Hong Kong; 8Faculty of Medicine and Health Sciences, Institute of Physiotherapy, The Jan Kochanowski University, 25-369 Kielce, Poland; 9Department of Psychology, CNRS UMR 6024, LaPSCo, University Clermont Auvergne, 63000 Clermont-Ferrand, France

**Keywords:** alternative medicine, mental health, occupation, prevention, stress

## Abstract

Introduction: Stress at work and psychosocial risks are a major public health problem. Sophrology and neurolinguistic programming (NLP) have demonstrated benefits in terms of mental, physical and social health, both in the general population and in patients, and both in and out of hospital settings. However, these approaches have never been provided at the hospital for the benefit of health professionals at risk of suffering at work. In general, we aim to demonstrate the effectiveness of a hospital sophrology/NLP intervention for health care professionals at risk of stress-related disorders. The secondary objectives are to study (i) within-group, and (ii) between-group): (1) effects on mental, physical, and social health; (2) persistence of effect; (3) relationships between job perception and mental, physical, and social health; (4) intervention success factors (personality and job perception, attendance and practice, other); (5) effects on other stress biomarkers (other measures of autonomic nervous system activity, DHEAS, cortisol, etc.). Methods: Our study will be a randomized controlled prospective study (research involving the human person of type 2). The study will be proposed to any health-care workers (HCW) or any non-HCW (NHCW) from a healthcare institution (such as CHU of Clermont-Ferrand, other hospitals, clinics, retirement homes). Participants will benefit from NLP and sophrology interventions at the hospital. For both groups: (i) heart rate variability, skin conductance and saliva biomarkers will be assessed once a week during the intervention period (6 to 8 sophrology sessions) and once by month for the rest of the time; (ii) the short questionnaire will be collected once a week during the whole protocol (1–2 min); (iii) the long questionnaire will be assessed only 5 times: at baseline (M0), month 1 (M1), month 3 (M3), month 5 (M5) and end of the protocol (M7). Ethics and dissemination: The protocol, information and consent form had received the favorable opinion from the Ethics Committee. Notification of the approval of the Ethics Committee was sent to the study sponsor and the competent authority (ANSM). The study is registered in ClinicalTrials.gov under the identification number NCT05425511 after the French Ethics Committee’s approval. The results will be reported according to the CONSORT guidelines. Strengths and limitations of this study: The psychological questionnaires in this study are self-assessed. It is also possible that responses suffer from variation. For the study, participants need to attend 6 to 8 sophrology sessions and one visit per month for 7 months, which might seem demanding. Therefore, to make sure that participants will complete the protocol, two persons will be fully in charge of the participants’ follow-up.

## 1. Introduction

Work stress and psychosocial risks are a major public health problem [1,2,3]. Health care workers (HCW) are particularly at risk [4,5,6,7,8,9,10]. Moreover, non-HCW (NHCW) staff of hospitals can also be at risk [11,12], with working environment described as the main risk factor [9,13]. The University Hospital (CHU) of Clermont-Ferrand is one of the three CHU in France with more than 10% absenteeism, and the first CHU for duration of sick leave. The political orientations favor an approach to improve the quality of life at work. Sophrology is a mind–body method used as a therapeutic technique or experienced as a philosophy of life. It is an exclusively verbal and non-tactile method. It combines a set of techniques that will act both on the body and on the mind through breathing exercises, muscle relaxation and mental imagery (or visualization). It allows us to find a state of well-being and to activate all physical and psychological potential in order to acquire a better knowledge of oneself [14]. Neurolinguistic programming (NLP) is a set of communication and self-transformation techniques that focuses on reactions rather than the origins of behaviors [15]. NLP associated with sophrology proposes above all to mobilize the resources of the unconscious. Although the level of evidence varied between studies (observational studies or randomized controlled trials with limited quality) [16,17], some research tend to support the effectiveness of NLP strategies in the improvement of mental health (anxiety and depression), physical, and social health in the general population [18], in individuals with social/psychological problems [17], and in patients [19,20,21]. For sophrology intervention in patients, the number of studies available is scarce despite some benefits reported in literature [14,22]. Additionally, these approaches have never been provided in the hospital for the benefits of workers at risk of stress-related disorders.

In general, we aim to demonstrate the effectiveness of a hospital sophrology/NLP intervention for HCW at risk of stress-related disorders. Stress and well-being will be the target of the intervention and will be measured both in a subjective (questionnaire) and objective (biomarkers) way. The main objective is to demonstrate an improved heart rate variability in the intervention group during the period between month 1 (M1) and M3 (sophrology sessions) of the intervention group in comparison with the same control period M1 to M3 of the “deferred intervention” group. The secondary objectives are to study for the following comparisons ((i) within groups: comparisons between and M0 to M7 by randomization group and (ii) between groups: comparisons at M5 and then between M0 and M7): (1) the effects of the intervention on mental, physical, and social health; (2) the long-term effects of the intervention; (3) the relationships between the perception of work and mental, physical, and social health; (4) the factors of success of the intervention (personality and perception of work, attendance rate, quality of personal practice, etc.); and (5) the effects on other biomarkers of stress such as other measures of the activity of the autonomic nervous system, DHEAS, cortisol, etc.

## 2. Materials and Methods

### 2.1. Study Design

Our study will be a randomized controlled prospective study (research involving the human person of type 2). The study design is described in Figure 1. The duration of participation per individual is 7 months for both the intervention and the control group. Randomization will be computer-generated.

### 2.2. Study Settings

We will recruit any HCW or any NHCW from a healthcare institution (such as CHU of Clermont-Ferrand, other hospitals, clinics, retirement homes, EHPAD, etc.), who will be randomized in two groups: (i) the intervention group will have a longitudinal follow-up of seven months in total (a control period of 1 month, then 6 to 8 sophrology sessions during two months, then a 4-month follow-up); (ii) the control group will have a differed intervention (longitudinal follow-up of seven months in total beginning by a control period without intervention of 3 months, then 6 to 8 sophrology sessions during two months, then a 2-month follow-up).

#### 2.2.1. The Intervention: Neurolinguistic Programming and Sophrology

Each participant will benefit from an NLP and sophrology intervention. The sophrology/NPL program will be provided by a qualified sophrologist outside from the hospital and that will be specifically employed by Clermont-Ferrand University Hospital for the purpose of this study, in order to avoid any previous relation between volunteers and the therapist. NLP and sophrology will take place within the University Hospital of Clermont-Ferrand. A total of 6 to 8 sessions will be proposed for each participant, by group of up to fifteen participants. Two sophrology sessions will be offered per week at two different times for easier accessibility. Each session will last one hour. They can take place in a seated or standing position, without physical contact. The sessions will consist of breathing exercises, visualization, and dynamic relaxation as well as concentration exercises. Breathing exercises have two objectives: (1) to manage the stress using deep inspiration and short expiration; (2) to dynamize the body using pauses between inspiration and expiration. During visualization process, we will ask participants to think of something or someone reminding pleasure and sensation of fullness. Dynamic relaxation includes exercises mobilizing parts of the body in order to take consciousness of all body segments. Breathing exercises, visualization, and dynamic relaxation are tools that should lead to an improvement of the dimensions of concentration (physical, mental, and emotional). All exercises will be guided by sophrologist. The sophrology/NLP program is standardized. Each session will take place in a typical way: (i) dialogue between the sophrologist and the participants, which makes it possible to specify the objective(s) of the session as the sophrologist will ask more specific questions to adapt the content of the sophrology session that they propose if they think it will be useful; (ii) presentation of the conduct of the session (possible variations) and instructions promoting autonomy; (iii) this practice is guided by the sophrologist, with the goal being to help free individuals from tensions, to play down a situation that worries and finally to realize that we can deal with. Sophrology exercises are adapted to the difficulties encountered by the participants; and (iv) time for free expression after the practice to promote the integration of the technique while giving elements to the sophrologist to guide the rest of the sessions.

#### 2.2.2. Description of Logistics Organization of the Study

Participants will benefit from NLP and sophrology interventions at the hospital. All measures will be conducted at the CHU Clermont-Ferrand. During the study, if a participant expresses particular psychic difficulties, support will be offered to him: appointment with the psychologist of the occupational health service of the CHU or referral to the attending physician according to his wishes.

#### 2.2.3. Biological Assessment

Each participant will perform a self-collection of saliva upon waking up, to control the circadian rhythm of biomarkers. Saliva sampling will be collected with the use of dedicated eppendorfs, at the CHU of Clermont-Ferrand. Centrifugation, aliquotage, and conservation of saliva will be achieved at the “Institut de Médecine du Travail, Faculté de Médecine, 28 place Henri Dunant, 63000 Clermont-Ferrand”. The following biomarkers will also be measured at the “Institut de Médecine du Travail”, using ELISA kits: Cortisol [23,24,25], DHEAS [23,24,26], Leptin [27,28], Ghrelin [28,29,30,31]. The samples will be kept until the end of the study and then they will be destroyed.

### 2.3. Ethics and Dissemination

The protocol, information and consent form had received the favorable opinion of the Ethics Committee North-West IV. The study is registered in ClinicalTrials.gov under the identification number NCT05425511. Notification of the approval of the Ethics Committee was sent to the study sponsor and the competent authority (ANSM). The results of questionnaires will be coded and managed through REDCap^®^ secure web platform with pseudonymous codification. The results will be reported according to the CONSORT guidelines.

### 2.4. Eligibility Criteria

The inclusion criteria are to be an HCW or an NHCW from a healthcare institution (such as CHU of Clermont-Ferrand, other hospitals, clinics, retirement homes), between 18 and 65 years, with a stress greater than or equal to 50 on a visual analogue scale (VAS) of stress, to be able to give an informed consent to participate in research and to be affiliated with a Social Security scheme. The exclusion criteria are to have psychiatric, cardiovascular (heart failure, arrhythmia, etc.), hepatic (liver failure, etc.), renal (kidney failure, etc.), or endocrinological diseases (diabetes, etc.) judged by the investigator to be incompatible with the study because they may interfere with the measurements. Persons who are non-affiliated to a health insurance or protected persons (minors, pregnant women, breastfeeding women, guardianship, curatorship, deprived of freedoms, safeguard of justice) or persons who refuse to participate are not included.

### 2.5. Recruitment Process

The study will be first proposed to workers of the University Hospital of Clermont-Ferrand (CHU)) using the mailing list of the staff. If there are not enough volunteers, then the study will be expanded to include all healthcare professionals. Volunteers will send an email with their contact to the address so-well@chu-clermontferrand.fr. A Clinical Research Assistant will contact them back and give them a first quick explanation of the study by phone, and will send them the information letter, so they can read it prior to the inclusion visit. Investigators will undergo the inclusion visits, either physically or by teleconsultation. Investigators will explain to each volunteer, in an exhaustive, clear and adapted manner, the research procedure, the potential benefit and the adverse effects taking into account their specificity (state of health, age, profession, habits and life projects, family environment, etc.) and will answer all questions. Participants can sign the consent form either during the consultation or later after a reflection period of 8 days. In this case, or for teleconsultation, the two signed consent forms will be returned by post or internal mail to the investigator. The investigator will keep one form, and the second form with both signature (investigator and volunteer) will be given to the volunteer at the baseline measure. They will be informed of the possibility, once their consent has been given, to withdraw it at any time.

### 2.6. Outcomes

The main outcome is heart rate variability change between M1 and M3. HRV will be measured using Zephyr™ BioHarness™ BT (Zephyr Technology, Annapolis, MD, USA). The main secondary outcomes are subjective (questionnaires: perceived stress, personality and perception of work, etc.) and objective markers of stress (biomarkers of stress such as other measures of the activity of the autonomic nervous system, DHEAS, cortisol, etc.). All outcomes are presented in Table 1.

For both groups (interventional and control groups): (i) heart rate variability, skin conductance and saliva biomarkers will be assessed once by week during the intervention period (6 to 8 sophrology sessions) and once by month the rest of the time; (ii) the short questionnaire will be collected once a week during the whole protocol (1–2 min); and (iii) the long questionnaire will be assessed only 5 times: at baseline (M0), month 1 (M1), month 3 (M3), month 5 (M5), and the end of the protocol (M7).

Outcomes will be collected by a research assistant without any link with the randomization process nor the intervention. The independence is necessary to minimize contamination of the intervention group, particularly in study designs which cannot be blinded.

## 3. Results

### 3.1. Justification of the Sample Size

This study aims to demonstrate an improved heart rate variability in the intervention group, in comparison with the control group. According to previous data [32], it seems reasonable to estimate that effect size, for the primary analysis between M1 and M3, will be around 0.5 for HRV parameters. Effect sizes are considered small between 0.20 and 0.50, moderate between 0.50 and 0.80, and high above 0.80 [33]. To highlight such clinically relevant difference, 85 participants per randomized group will be needed for a two-sided type I error at 5% (alpha level, i.e., showing a difference that does not exist), a statistical power at 90%. If the desired power level is typically 80%, we specified a higher power level of 90%, so there would be a 90% probability that we will not commit a type II error (beta-level, i.e., failing to show a real difference) [34,35]. Finally, it is proposed to include 100 patients per group to consider loss to follow-up.

### 3.2. Data Collection and Management

The material is presented in Table 1.

**Table 1 ijerph-20-01185-t001:** Synthesis of primary and secondary outcomes.

Variables	Type of Measures	Modalities to Measure	Time of Measures		References
			Group Intervention	Group “Waiting List”	
Autonomic nervous system	Heart rate variability	Zephyr™ BioHarness™ BT	**Intervention**: 1 during each sophrology sessions (6 to 8 sessions) **Rest of the time**: 1 by month		[36]
	Skin conductance	Wristband electrodes—Empatica E4, Milano, Italy	**Intervention**: 1 during each sophrology sessions (6 to 8 sessions) **Rest of the time**: 1 by month		[37]
Demographic *	Age, gender, marital status, children, occupation, and recent stressful event	Questionnaire	**Age, gender and recent stressful event**: once at inclusion then every week **Marital status, children and occupation**: once at inclusion		[36]
Clinical measurements	Height, weight	Questionnaire	**Height**: once at inclusion **Weight**: 5 times (M0, M1, M3, M5 and M7)		[38]
Psychology and quality of life	Depression	HAD 7 items	**5 times**: M0, M1, M3, M5 and M7		[39]
	Anxiety	HAD 7 items			[39]
	Burn-out	Maslach Burn-Out Inventory			[40]
	Perception of work	Job Content Questionnaire of Karasek			[27]
		Effort Reward Imbalance			[41,42,43]
	Alexithymia	Toronto Alexithymia Scale (TAS-20)			[44,45,46,47,48]
	Stress at home, fatigue, sleep quality, anxiety, mood, family support, stress at work, burnout, job control, job demand, hierarchy support, colleagues support, effort reward imbalance	Visual analog scale of 100 mm	1 by week during the whole protocol		[36]
Lifestyle	Coffee/tea, food intakes	Questionnaire	**Coffee/tea**: once by week during the whole protocol **Food intakes**: 5 times (M0, M1, M3, M5 and M7)		[11]
	Physical activity	Recent Physical Activity Questionnaire (RPAQ)	5 times (M0, M1, M3, M5 and M7)		[49]
		Questionnaire	1 by week during the whole protocol		
	Consumption of alcohol, tobacco, cannabis and medication	Questionnaire	**Alcohol, tobacco and cannabis**: every week **Medication**: 5 times (M0, M1, M3, M5 and M7)		[7]
Sophrology practice	Number of sessions per week	Questionnaire	1 by week during the whole protocol		
	Stress before and after the last sophrology session	Visual analog scale of 100 mm	1 by week during the whole protocol		[7]
Allostatic load **	Cortisol	Saliva sampling and deep-freezing	**Intervention**: 1 during each sophrology sessions (6 to 8 sessions) **Rest of the time**: 1 by month		[23,24,25] (1, 2)
	DHEAS	Saliva sampling and deep-freezing			[23,24,26]
	Leptin	Saliva sampling and deep-freezing			[27,28]
	Ghrelin	Saliva sampling and deep-freezing			[28,29,30,31]

* Adjustment variables; ** Saliva samples upon waking up to control the circadian rhythm of biomarkers.

#### 3.2.1. Heart Rate Variability (HRV)

A Zephyr™ BioHarness™ BT (Zephyr Technology, Annapolis, MD, USA) will record HRV parameters. The Zephyr™ BioHarness™ BT is a heart rate transmitter belt simply positioned on the chest, with a 26 h recording time, a beat per minute within a 25–240 range, and respiratory rate within a 3–70 range. The aim will be to demonstrate that is it possible to detect anxiety using this easy to use and cheap device. Anxiety could also be detect using the respiratory rate. In addition, the Zephyr™ BioHarness™ BT will also measure the trunk’s movements with a 3-Axis accelerometer. Recommendations of the European Society of Cardiology and the North American Society (Task Force) will be followed to examine the HRV data. Time and frequency domains will be used to explore HRV. [50]. We will also apply the methodology developed by our team [51].

We will visually check the premature beats who will be automatically reject. Several element will be analyzed in time domain: R–R intervals and their standard deviation, the number of adjacent N–N differing by more than 50 ms divided by the total number of N–N inter-vals (pNN50) and the square root of the mean squared difference of successive R–R intervals (rMSSD). The pNN5 and rMSSD are related with high frequency (HF) power and therefore to parasympathetic activity. High (0.15–0.4 Hz) and low (0.04–0.15 Hz) frequency power will be examine in the spectral domain. HF is the principal efferent parasympathetic (vagal) activity to the sinus mode, LF is an indicator for both parasympathetic and sympathetic functions. The variations of thermoregulatory mechanisms, activity of the renin–angiotensin system and the function of peripheral chemoreceptors are partially reflects by very low frequency (VLF, 0.003–0.04 Hz). Normalized units (nu), corresponding to the relative value of each power component in proportion to the total power minus the VLF component, will also be used to assessed LF and HF. Thus, the best sympathetic and parasympathetic activity seems to be represented by LFnu and HFnu, respectively. The LF/HF ratio, that is, the sympathovagal balance, will also be calculated (Table 2) [52].

#### 3.2.2. Skin Conductance

Skin conductance will be measured in micro-Siemens with sampling rates at 2, 4, 8, 16 and 32 Hz during the phases 1 to 3. The skin conductance sensor (Empatica E4, Milano, Italy) is set on a wristband [53]. 

#### 3.2.3. Questionnaires

The REDCap^®^ questionnaires in digital format will be automatically emailed to participants. They will then have to click on a link to answer. Only the short one will be sent to the participants. The long questionnaire will be completed during monthly appointments (M0, M1, M3, M5 and M7).

The short questionnaire will assess sociodemographic such as gender, age and recent stressful event. There are visual analog scales (VAS) too such as stress at home, fatigue, sleep quality, anxiety, mood, family support, stress at work, burnout, job control, job demand, hierarchy support, colleagues support, effort reward imbalance on a horizontal, non-calibrated line of 100 mm, ranging from very low (0) to very high (100) [36,54]. Lifestyle will be assessed in short questionnaire: physical activity, addiction (alcohol, cannabis, and tobacco) and coffee/tea. Sophrology practice will be assessed too: number of sessions per week, stress before and after the last sophrology session.

The long one will contain the following questionnaires. The Recent Physical Activity Questionnaire (RPAQ) [49] is designed to assess the level of physical activity and sedentary behavior in adults over the past four weeks. It is divided into three parts: work and study, home and leisure, and stairs and travel. It classifies individuals into several categories, one based on sedentary time (low; moderate, i.e., 3–7 h/day; or high, i.e., >7 h/day) and one based on physical activity level (<8.3 MET.h/wk: inactive or >8.3 MET.h/wk: active). The Hospital Anxiety and Depression scale (HAD) [55] evaluate the frequency of anxious (HAD-A: 7 items) and depressive (HAD-D: 7 items) symptoms using a 14-items questionnaire. For each item, four frequency choices are offered ranging from “never” to “always”. For the two components (HAD-A and HAD-D), scores make it possible to determine two levels of symptomatology: “possible symptomatology” (from 8 to 11) and “clear and proper diagnosis of depressive or anxiety disorder” (>11). The Maslach Burn-out Inventory (MBI) [40] is designed to assess the three components of the burn-out syndrome: depersonalization (five items), emotional exhaustion (nine items), and reduced personal accomplishment (eight items). It is composed of 22 items related to personal feelings or attitudes. Items are made of a seven frequency choices, varying from “never” to “every day”. A score, which can be split into three categories (low, average, or high) is determine for each components separately but there are not merge to obtain a global score. The Job Demand-Control-Support (JDSC) questionnaire of Karasek [27] is a twenty-six-item questionnaire decomposed into nine items to evaluate job demand, nine items to evaluate job control and eight items to measure social support. The items are made of a four-point scale measuring level of agreement, ranging from “strongly disagree” to “strongly agree”. A reverse scoring is required for 5 of the 26 items. According to French data, the job demands is considered as high for a score higher than 20 and the job control and social support as low for score lower than 71 and 24, respectively. High demand and low control characterize jobstrain. Jobstrain associated with low social support reflects the isostrain situation. We would ask the participants to fulfil the questionnaire from memories that they keep of their work. The Effort-Reward Imbalance Questionnaire (ERI) is a self-administrated test assessing psychological distress and health problems that may occur when there is an imbalance between the efforts required by the work and the rewards received. We used the 23 items of the short version of the ERI model exploring efforts (six items), over commitment (six items), and rewards (eleven items) [43]. Items of ERI were scored on a five-point Likert-type scale, ranging from 1 = disagree to 5 = agree and very disturbed. A ratio extrinsic efforts and rewards can assess the imbalance between these two dimensions. A ratio greater than one defines employees exposed to an imbalance between efforts and rewards [43]. The Twenty-item Toronto Alexithymia Scale (TAS20) [44,45,46,47,48] is the most widely used self-report measure of alexithymia. Alexithymia (from the Greek, a = lack of, lexis = work or word, and thymos = mood or emotion) refers to severe reductions in both the cognitive and affective components of emotional experience. For this scale, a three-factor structure was proposed: difficulty identifying feelings, difficulty describing feelings and externally oriented thinking. Responses are made on a 5-point scale from “strongly agree” to “strongly disagree”. The long questionnaire also contains questions about food intake in the last month.

### 3.3. Data Analysis

The SO-WELL trial statistical analysis plan (and its successive versions) will be kept in the study records. The statistical analysis plan may be revised during the study, e.g., in order to take into account amendments to the protocol or any change in the conduct of the study that may have an impact on the statistical analysis plan as described in its current version.

Statistical analysis will be performed using Stata software (v16, Stata-Corp, College Station, TX, USA). According to the International Conference on Harmonization Good Clinical Practice guidelines, analyses will be conducted before the breaking of the randomization code. The primary analyses will be carried out in two stages. First, using intention-to-treat analysis and applying imputation data method for missing data. Then, using per-protocol analysis which consists of consider only participants without missing data for the primary endpoint. 

Categorical parameters will be described in terms of numbers and frequencies, whereas continuous variables will be expressed as mean and standard deviation or median and (inter-quartile range) according to statistical distribution. The assumption of normality will be studied using Shapiro–Wilk test. 

At inclusion, participants will be characterized and compared between groups according to the following variables: respect with eligibility criteria, epidemiological, clinical and treatment attributes. Using the characteristics of the participants and potential factors associated with the primary outcome, the comparability at baseline between groups will be assessed. According to both clinical and statistical considerations, a possible difference between the two groups on one of these characteristics will be determined. 

Intergroup comparisons will be systematically conducted (i) without adjustment and (ii) by adjusting for factors whose distribution could be unbalanced between groups. 

All statistical tests will be two-sided and *p* < 0.05 will be considered significant. Most of analyses of the secondary evaluation criteria will be exploratory. As discussed by Feise [39], the adjustment of the type I error will not be systematically proposed, but case by case in view of clinical considerations and not only statistical. 

#### 3.3.1. Analyses for the Primary Outcome

To compare the primary endpoint between groups, we will use Student’s t-test, or the non-parametric Mann–Whitney test if assumptions of t-test are not met. 

For the primary analysis, the intention-to-treat analysis will be considered, i.e., the comparisons between groups (“intervention” group and “waiting list” group) between M1 and M3. To prevent attrition bias, imputation of the missing data is planned. Then, the analysis of the primary outcome will be completed by multivariable analysis using a linear mixed model to compare HRV parameters between randomized groups considering (i) covariates identified according to clinical relevance (such as sex, age, number of complete sessions performed, occupational characteristics) and univariate results and (ii) session as random effects (to measure the variability between and within session). We will study the normality of the residuals from linear regression. If needed, a logarithmic transformation of the dependent variable will be suggested. Effect sizes and 95% confidence intervals will be used to express result. 

#### 3.3.2. Analyses for the Secondary Outcomes

Randomized groups comparison will be carried out (i) as aforementioned for continuous secondary endpoints and (ii) for categorical endpoints using Fisher exact tests or chi-squared. The results will be introduced using median difference and 95% confidence intervals estimated using quantile regression model for non-Gaussian data. The results will be presented using absolute differences and 95% confidence intervals for categorical parameters. The multivariable analysis related to dichotomous endpoints will be generalized linear mixed model, with logit link function, and center as random-effect. The results will be expressed with of relative risks and 95% confidence intervals.

Longitudinal data will be analyzed using random-effects regressions, modeling between and within participant effect, as random-effect, in addition to session effect. These analyses will concern (i) within groups comparisons between and M0 to M7 by randomization group and (ii) between groups: comparisons at M5 and then between M0 and M7). For between groups comparisons, the following fixed effects will be studied: randomization group, time-points evaluation and their interaction. 

Subgroup analyses will be conducted to explore potential influence of sex, occupation, stress at baseline, and previous sophrology experience on the primary outcome. Interaction terms in the regression models will be used to test for heterogeneity of effect between subgroups.

#### 3.3.3. Methods Considering Missing, Unused or Invalid Data

The statistical nature of missing data will be studied using sensitivity analysis. It will also be used to choose the most appropriate approach for the imputation of missing data: maximum bias (e.g., last observation carried forward, baseline observation carried forward) or estimation proposed by Verbeke and Molenberghs for repeated data. 

Particular attention will be paid to study of the adherence of participants to the intervention. Participants abandoning will be studied using this parameter as a censored data, which will involve the use of Kaplan–Meier to estimate it and log-rank for the comparison between groups. 

## 4. Discussion

The World Health Organization defines health as “not merely the absence of disease or infirmity but a state of complete physical, mental and social well-being” [56]. In fact, the role of well-being at work plays a major role in social well-being [57]. HCW are particularly at risk [4,5,6,7,8]. Moreover, NHCW staff of hospitals can also be at risk [11,12], with working environment described as the main risk factor [13]. Indeed, the medical staff, especially those working in emergency departments are exposed to psychosocial risk factors because of the type of work they do [58,59]. Moreover, the COVID-19 pandemic has worsened the situation all around the world [60,61,62]. In general, HCW are confronted with the death of patients and the distress of their families [63]. 

Sophrology is a mind–body method used as a therapeutic technique or experienced as a philosophy of life. It helps to regain a state of well-being and to activate all physical and psychological potentials in order to acquire a better knowledge of oneself [14]. NLP is a set of communication and self-transformation techniques that focuses on reactions rather than on the origins of behaviors [15]. NLP combined with sophrology proposes above all to mobilize the resources of the unconscious.

This protocol seems particularly interesting for the University Hospital (CHU) of Clermont-Ferrand because it is one of the three CHU in France with more than 10% absenteeism, and the first CHU for duration of sick leave. By proposing sophrology to professionals at risk of suffering at work, this study design will allow us to observe a short-term effect (the first 3 months) and a long-term effect (4 months follow-up). This study will provide important data about the benefits of sophrology and NLP for workers at risk of stress-related disorders.

Articles related to this study protocol will follow the CONSORT recommendations for reporting randomized trials.

### Potential Limitations

The psychological questionnaires in this study are self-assessed and may suffer from a bias of auto-evaluation. For the study, participants need to attend 6 to 8 sophrology sessions and one visit per month for 7 months, which might seem demanding. Therefore, to make sure that participants will complete the protocol, one person will oversee the participants’ follow-up. We are not immune to a confusion bias. For example, observed changes may be produced by other factors than NLP, such as organizational changes during the intervention period. However, we tried to minimize this risk. First of all, healthcare professionals or volunteers from healthcare institutions will be very diverse and will not belong to a single department. In addition, the RCT design will also limit the risk of confusion bias as participants will be both compared to a control group with deferred intervention (intergroup comparison) and their own control (intragroup comparison). Blinding assessment is impossible considering the nature of the intervention. However, outcomes will be collected by a research assistant independently from the randomization process or the intervention to minimize contamination of the intervention group.

## 5. Conclusions

Stress at work and psychosocial risks are a major public health problem. Not only HCW but also NHCW staff of hospitals can be at risk, with working environment described as the main risk factor. The political orientations favor an approach to improve the quality of life at work. For this, NLP associated with sophrology may improve the mental (anxiety and depression), physical, and social health. Finally, these approaches have never been provided in the workplace for the benefits of workers at risk of stress-related disorders.

## Figures and Tables

**Figure 1 ijerph-20-01185-f001:**
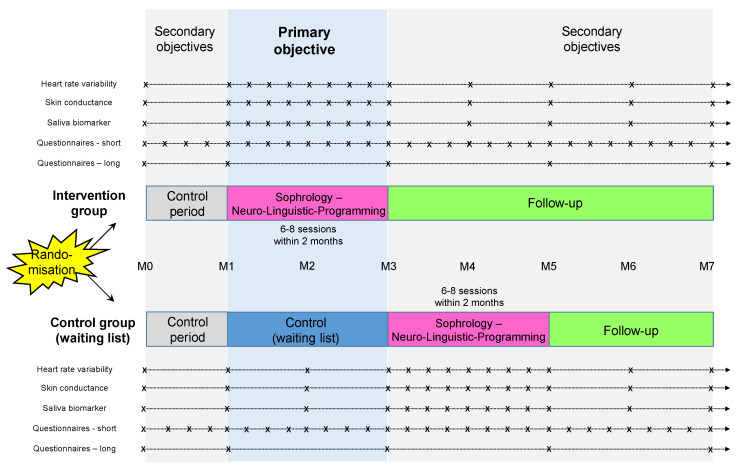
Study design. “×” for each time of measurement.

**Table 2 ijerph-20-01185-t002:** HRV components description.

Acronym	Full Name	Unit	Interpretation
**Time-domain**			
RR	RR–intervals (or Normal to Normal intervals—NN), i.e., beat-by-beat variations of heart rate	ms	Overall autonomic activity
SDNN	Standard deviation of RR intervals	ms	Correlated with LF power
rMSSD	Root mean square of successive RR-intervals differences	ms	Associated with HF power and hence parasympathetic activity
pNN50	Percentage of adjacent NN intervals varying by more than 50 milliseconds	%	Associated with HF power and hence parasympathetic activity
**Frequency-domain**			
TP	Total power, i.e., power of all spectral bands	ms^2^	Overall autonomic activity
LF	Power of the low-frequency band (0.04–0.15 Hz)	ms^2^ (absolute power) or nu (relative power in normalized unit)	Index of both sympathetic and parasympathetic activity, with a predominance of sympathetic
HF	Power of the high-frequency band (0.15–0.40 Hz)		Represents the most efferent vagal (parasympathetic) activity to the sinus node
LF/HF	LF/HF ratio	-	Sympathovagal balance
VLF	Very low frequency (0.003–0.04 Hz)	ms²	Sympathovagal balance

## Data Availability

Not applicable.

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
