# Peer review of "SOphrology Intervention to Improve WELL-Being in Hospital Staff (SO-WELL): Protocol for a Randomized Controlled Trial Study"

_ijerph, 2023, doi:10.3390/ijerph20021185_

Round 1

Reviewer 1 Report

This study protocol is intended to demonstrate the effectiveness of a hospital sophrology - neurolinguistic intervention for Hospital staff at risk of stress-related disorders. For this purpose stress and well-being/ mental health perception will be measured and monitorized with both subjective self informed questionnaires and objective biomarkers such as heart rate, skin conductance, and other biomarkers obtained by saliva sampling.

Study is well designed and protocol is sufficiently described although some aspects requiere further specifications:

- Therapist: experience, persons implementing treatment, manualization and structure of sessions... should be detailed.

- Control group: although waiting list is considered an adequate control group, reviewer suggests to use another control group with an active treatment component (placebo control group). See this review for further recommendations: https://pubmed.ncbi.nlm.nih.gov/35377466/

- Regarding references, although appropriate, there is a large amount of evidence published prior to 2017, so I would suggest to search for up to date recent references in the field.

I have no other suggestions to make.

Reviewer 2 Report

This manuscript is the study protocol of a planned randomized controlled trial (RCT) to investigate the effectivness of SOphrology / neurolinguistic programming (NLP) on work stress and health-related outcomes among health care workers. Well-tested questionnaires to measure work stress and saliva biomarkers of strain reactions will be applied to investigate the intervention effects.

Major points:

Evidence for NLP is not as clear as stated by the authors. A systematic review of experimental studies including RCTs did not reveal a substancial effect of NLP on health-related outcomes (Sturt et al. Br J Gen Pract. 2012; 62: e757-64). The metanalysis by Zaharia et al. 2015 quoted by the authors is based on observational studies only.

The intervention concentrates on behavioral and cognitive / motivational changes induced by NLP which should improve work-related stress and strain reactions. How can be made sure that observed changes are not produced by other factors than NLP, e.g. organisational changes during the intervention period?

I could not see whether this protocol is following any established report form for RCTs like CONSORT.

It is not clear whether randomisation and intervention are independent from the evaluation. The independence is necessary to minimize contamination of the intervention group, particularly in study designs which cannot be blinded.

A simple bivariate test of the primary outcomes ( lines 332 ff.) is not satisfying. Primary outcomes may vary between sub groups of the intervention population, e.g. according to the type of ward. A multivariat analysis considering between and within subgroup differences is needed.

Minor points:

There is a mistake in line 289. The French version of ERI does not consist of 46 items.

It would be helpfull to have a bit more information about how the power analysis was conducted.

Line 209: loss to follow-up instead of lost to follow-up.
